# How Does the Context Shape the Technical Support from the Provincial Health Administration to District Health Management Teams in the Democratic Republic of Congo? A Realist Evaluation

**DOI:** 10.3390/ijerph21121646

**Published:** 2024-12-10

**Authors:** Samuel Bosongo, Zakaria Belrhiti, Faustin Chenge, Bart Criel, Bruno Marchal, Yves Coppieters

**Affiliations:** 1Faculté de Médecine et Pharmacie, Université de Kisangani, Kisangani 2012, Democratic Republic of the Congo; faustin.chenge@unilu.ac.cd; 2École de Santé Publique, Université Libre de Bruxelles, 1070 Bruxelles, Belgium; yves.coppieters@ulb.be; 3Institute of Tropical Medicine Antwerp, 2000 Antwerp, Belgium; bcriel@ext.itg.be (B.C.); bmarchal@itg.be (B.M.); 4Centre de Connaissances en Santé en République Démocratique du Congo, Kinshasa 3088, Democratic Republic of the Congo; 5Département de Santé Publique and Management, École Internationale de Santé Publique, Université Mohammed VI des Sciences de la Santé, Casablanca 82403, Morocco; zbelrhiti@um6ss.ma; 6Centre Mohammed VI de la Recherche et Innovation (CM6RI), Rabat 10112, Morocco; 7École de Santé Publique, Faculté de Médecine, Université de Lubumbashi, Lubumbashi 1825, Democratic Republic of the Congo

**Keywords:** technical support, district health management teams, provincial health administration, Democratic Republic of Congo, realist evaluation

## Abstract

Since 2014, the health sector in the Democratic Republic of the Congo has been undergoing reforms aimed at strengthening the Provincial Health Administration (PHA) to better support health district development through technical support to district health management teams (DHMTs). However, there is limited understanding of how, for whom, and under what conditions this support works. Using a realist evaluation approach, this study aimed to test an initial program theory of technical support to DHMTs by PHA staff in Kasai Central Province. Data were collected from document reviews, interviews, questionnaires, and routine health information systems. After thematically analysing the implementation, context, actors, mechanisms, and outcomes, we applied retroductive reasoning to connect these elements using the Intervention–Context–Actors–Mechanisms–Outcomes configurations (ICAMOcs) heuristic. We identified nine ICAMOcs showing how resource constraints and political and organisational challenges hindered the effective delivery of technical support. These challenges triggered disabling mechanisms, such as low motivation, self-efficacy, a sense of accountability, psychological safety, reflexivity, the perceived relevance of support, the perceived credibility of PHA staff, and perceived autonomy, resulting in mixed outcomes. The performance-based financing scheme helped mitigate some issues by providing resources and boosting extrinsic motivation, but concerns persist about its sustainability due to reliance on external funding. These findings highlight the need for strong political commitment and coordinated efforts to address these challenges.

## 1. Introduction

Achieving Sustainable Development Goal 3 (SDG3) related to health and well-being, particularly the universal health coverage (UHC) target, requires strong health systems capable of effective stewardship, resource creation, financing, and service delivery [1,2]. However, progress toward SDG3 is hindered by weak health systems, especially in low- and middle-income countries like the Democratic Republic of the Congo (DRC) [2].

The DRC has one of the worst-performing health systems in Africa [1,3,4]. While there has been some progress toward achieving SDG3 [5], health indicators remain alarming. Life expectancy dropped from 60 years in 2019 to 59 in 2021 [6]. The maternal mortality ratio decreased from 846 per 100,000 live births in 2014 [7] to 547 in 2020 [6], but the DRC still ranks 9th highest in Africa [8]. Similarly, under-five mortality dropped from 104 per 1000 live births in 2014 to 92 in 2024 [9].

The prevalence of HIV/AIDS dropped from 1.2% in 2014 [7] to 0.7% in 2023 [10]. The DRC ranks second globally for malaria cases [11] and eighth for tuberculosis [12]. In addition to these three diseases, which account for 19% of the country’s total Disability-Adjusted Life Years (DALYs) [13], the DRC faces public health emergencies, including outbreaks (Ebola Virus Disease, COVID-19, Mpox, etc.), chronic armed conflicts, and consequences of climate change such as floods. The country’s health system is not resilient enough to prepare for, effectively respond to, learn from, and recover from these crises [14].

In addition to infectious diseases, which account for 25,688.48 DALYs per 100,000 (54% of the country’s total DALYs [13]), non-communicable diseases like diabetes, cardiovascular diseases, chronic respiratory diseases, and cancers are rising [14], accounting for 15,123.43 DALYs per 100,000 (32% of the country’s total DALYs) [13]. The recent demographic and health survey revealed that the prevalence of arterial hypertension was 9% among women and 15% among men aged 15 to 49 years. The prevalence of diabetes was 4% in both sexes and age groups. The readiness of health facilities to deliver non-communicable disease services remains low [15,16].

This concerning situation affects people differently depending on their socio-demographic and economic characteristics and their location. Unfortunately, social determinants of health are not systematically addressed, making poor people and those living in remote rural areas, slums, and conflict areas the most affected.

To address this situation and achieve SDG3, a stronger health system is required, as emphasized by the Alliance for Health Policy and System Research [2]. In response to this call, the Ministry of Health (MoH) developed the Health System Strengthening Strategy in 2006 [17], which was updated in 2010 [18], in line with four reforms proposed by the World Health Organization in its 2008 World Health Report [19]. The Health System Strengthening Strategy reaffirmed the importance of primary health care and emphasised the development of Health Districts (HDs). The reform of the Provincial Health Administration (PHA) was recommended in order to effectively support this development [18]. This reform involved separating inspection/control from support functions, restructuring the PHA office from thirteen to six departments, and gradually shifting from bureaucratic to adhocratic functioning [20]. PHA staff, known as *“encadreurs provinciaux polyvalents”* in French, were assigned to provide technical support to District Health Management Teams (DHMTs) through supervision and coaching in order to enhance their management capacities so that they could better steer the development of their HDs.

To achieve this goal, technical support needs to incorporate key features of successful capacity-building programs for health district managers. These include adopting an action-learning (or learning-by-doing) approach, being team-based, flexible, and adaptable, and fostering supportive interactions between facilitators and participants [21]. Such an approach aligns with established adult learning theories, including Kolb’s experiential learning theory [22], Knowles’ adult learning theory [23], and Mezirow’s transformative learning theory [24]. Kolb’s theory posits that concrete experiences lead to reflective observation, from which abstract concepts are developed and tested in new experiences [22]. Knowles’ theory underscores the importance of self-directed learning, leveraging prior experiences (including errors), addressing relevant and practical problems, and fostering intrinsic motivation [23]. Mezirow’s theory focuses on the critical reflection of values, beliefs, and assumptions, enabling individuals to adopt new, meaningful perspectives [24]. By integrating these theories into technical support processes, capacity building efforts can promote experiential, self-directed, and transformative learning tailored to the specific needs and context of DHMT members.

Between 2014 and 2015, the PHA reform was rolled out across the 26 provinces of the DRC, with PHA staff providing technical support to DHMTs. However, little is known about how the context shapes the implementation process and the outcomes of this support because of the scarcity of studies. Existing studies [25,26] have not explored why, how, or under what conditions the technical support from the PHA staff to DHMT members works or not. To address this gap, a realist evaluation [27] was proposed as part of the first author’s PhD research. The study began with developing an Initial Program Theory (IPT), hypothesising how mechanisms activated by the intervention through relevant actors in specific contexts could generate expected outcomes [21]. This paper details the process of testing the IPT in the province of Kasai Central, in order to refine it based on empirical findings. The refined theory will inform policymakers and health managers on the actual functioning of technical support to HDs and offer recommendations for its refinement, adaptation, or improvement [28].

## 2. Materials and Methods

### 2.1. Study Approach: Realist Evaluation

Realist evaluation (RE) is a theory-driven approach that seeks to explain why, how, for whom, and under what circumstances programs work or not [29]. RE is rooted in scientific realism, which posits that reality exists independently of researchers (positivist ontology), but the understanding of this reality is influenced by the researcher’s perspective (relativist epistemology) [30]. It emphasises that change is not caused by interventions themselves but by how people respond to the interventions’ resources or opportunities (mechanisms) within their specific contexts. Thus, identifying these mechanisms is central to RE.

Realist researchers view social programs as active theories embedded within social systems, requiring them to elicit, test, and refine underlying program theories [31]. A program theory explains how people’s responses to program resources (mechanisms) lead to observed outcomes in specific contexts. Technical support, as a capacity-building intervention, is a complex (social) program. It involves multiple interacting actors from various social sub-systems with differing and sometimes conflicting values, norms, powers, and expectations [32,33]. RE is well-suited for studying complex social programs as it helps to understand generative causation. Realists believe that objects have intrinsic properties or powers (mechanisms) that generate certain outcomes, and the exercise of these powers is contingent on specific conditions. Thus, the same causal mechanisms can produce different outcomes under different conditions, or different mechanisms can produce the same outcomes [34].

In practice, RE follows a three-stage iterative process: eliciting, testing, and refining (initial) program theory. Researchers use the context–mechanism–outcome configuration (CMOc) as a heuristic tool. In this study, we used a variant known as the intervention–context–actors–mechanisms–outcomes configuration (ICAMOc) to distinguish intervention from context and emphasize the role of actors [35]. De Weger et al. [36] suggest that incorporating additional explanatory components into CMO configurations can be valuable, provided the configuration follows the principle of generative causation.

### 2.2. Eliciting the Initial Program Theory

We developed the IPT using a scoping review [37], a policy document review, and interviews with program designers [21]. The IPT is structured across three levels: the PHA office, the PHA–DHMT interface, and the district level, reflecting the long implementation chain of technical support. Hypothesizing IPT at each level is crucial, with outcomes from higher levels affecting lower levels (ripple effect, [38]). Details on eliciting the IPT are in a previous paper [21], and the IPT is outlined in Table 1.

### 2.3. Study Design and Setting

We used a single case study design, which is relevant for inquiring about complex phenomena like technical support in real-life contexts [39]. This design benefits from developing theoretical propositions beforehand to guide data collection and analysis, aligning with this study’s aim of testing and refining an a priori IPT [39,40,41]. The case was defined as technical support provided by PHA staff to DHMT members, with the PHA as the unit of analysis and two HDs as sub-units.

We selected Kasai Central province based on criteria defined in the study protocol [27], including 2021 performance ranking (the highest, Appendix A), funder presence (i.e., World Bank), and geographic location in the country (centre). Accessibility for the first author, who is pursuing a PhD, was also considered. Within the province, two HDs, Katoka (urban) and Bunkonde (rural), were chosen based on location. Key features of the province and selected HDs are outlined in Table 2.

### 2.4. Data Collection Methods and Tools

RE is method neutral. Depending on the IPT to be tested, both quantitative and qualitative methods are used [30]. Quantitative data are often collected to measure the outcomes, while qualitative methods allow for exploring the underlying mechanisms and describing contextual conditions as well as the implementation process. We collected data using various qualitative and quantitative methods and tools, such as document review, semi-structured interviews, questionnaires, and routine data from DHIS2 (District Health Information System, version 2). Qualitative data were gathered through semi-structured interviews and document reviews, focusing on intervention implementation, contextual factors, mechanisms, and perceived outcomes. Quantitative data, collected via questionnaires, routine data, and document reviews, primarily addressed outcomes and DHMT members’ perceptions of the technical support process and its results. Table 3 details the purposes, tools, participants, sampling type, and size for each method.

### 2.5. Data Analysis

We followed a two-step iterative process for data analysis, which involved identifying ICAMO components and formulating ICAMO configurations (Figure 1).

#### 2.5.1. Step 1: Identifying ICAMO Components

The first step involved the individual analysis of quantitative and qualitative data and categorising them into the ICAMO components, i.e., intervention, context, actors, mechanisms, and outcomes (defined in Table 4).

In the quantitative strand, a descriptive analysis using Microsoft Excel was performed in order to assess the management capacity of DHMTs and the performance of their HDs. The assessment of the management capacity of DHMTs covered both individual management functions and overall capacity. The evaluated functions included coordination, planning, monitoring and evaluation, hands-on training, supportive supervision, health information management, epidemiological surveillance, resource management (human, financial, material, and pharmaceutical), and research. Each function was scored on a scale from 1 (very poor) to 4 (very good) using predefined criteria in the scorecard. Management capacity for each function was expressed as a percentage, calculated by dividing the obtained score by the maximum possible score for that function (which varied by the number of items assessed). Overall management capacity was similarly calculated as a percentage of the total score out of a maximum of 112 (see Appendix A for details). The management capacity of DHMTs was categorised into three levels: good (80–100%), average (50–79.9%), and low (<50%). These thresholds were defined based on the national ranking system for PHAs and HDs.

Health district performance was measured using ten key performance indicators (KPIs) derived from the monitoring and evaluation framework of the National Health Development Plan (NHDP) 2019–2022. These KPIs served as proxies for assessing the access, quality, and equity of healthcare services. The indicators included overall completeness rate, curative service use rate, antenatal care (4 visits) coverage, skilled birth attendance, contraceptive prevalence, pentavalent vaccine (DPT-HepB-Hib3) coverage, in-hospital mortality rate (>48 h), tuberculosis treatment success rate, percentage of HIV-positive pregnant women on antiretroviral therapy, and percentage of children under five with malaria treated per national protocol. Each KPI was assigned a target from the NHDP framework and scored from 0 (very poor) to 4 (very good) based on performance relative to the target. The overall performance for each year was calculated as a percentage: the total score obtained by the health district divided by the maximum possible score (40). HD performance was classified into three categories: good (80–100%), average (50–79.9%), and low (<50%), consistent with the classification used for the management capacity of DHMTs. More details on the calculation of HD performance can be found in Appendix A.

In the qualitative strand, data were managed with N-Vivo 14 [42]. The first author conducted a deductive thematic analysis of interview transcripts, document review synthesis, observation notes, and field notes using a codebook developed on the basis of the IPT (Table 4) while paying attention to emerging new themes.

**Table 4 ijerph-21-01646-t004:** Codebook.

ICAMO Elements		Definitions	Coding Rules
Intervention		A combination of policy or program components or strategies, especially those meant to change people’s behavioural [43].	Use this code to document any technical support features (content, approaches, and intensity) used to achieve the expected outcomes.
Context		Any pre-existing social, economic, cultural, political, or other environmental factor that may influence the implementation and/or the actors, and that may shape the outcomes.	Use this code to document social, economic, cultural, political, organizational, or other environmental factors that enable or hinder the expected outcomes.
Actors		The people, groups, and institutions who are addressed by the intervention and who are central to its adoption and implementation.	Use this code to capture any attributes (background, experience, knowledge, skills, and attitude), actions, or actual practice of an individual, group, or institution.
Mechanisms		People’s reasoning and reactions to resources made available by the intervention, triggered in specific contexts [44].	Use this code to capture why actors behave or act to achieve or not achieve the expected outcomes.
Outcomes	Short-term	The immediate effect of program activities [43].	Use this code to document changes in the intervention’s direct beneficiaries’ knowledge, skills, or awareness.
	Mid-term	Behavioural changes that follow the immediate knowledge and awareness changes [43].	Use this code to capture the changes that follow the changes in knowledge, skills, or awareness, such as changes in district performance.
	Long-term	Changes over the long term, such as health status and impact on community and health system [43].	Use this code to document the further indirect impact of the intervention.

#### 2.5.2. Step 2: Formulating ICAMO Configurations

In the second step, we connected individual ICAMO components identified in Step 1 using a retroductive approach. Retroduction as used in the realist approach is focusing on mechanism-centred inference: it aims at uncovering the causal mechanisms behind observed outcomes [40,45,46]. We described the observed outcomes, explored the mechanisms linking them to the intervention features, and examined the contextual factors that facilitate or impede these mechanisms. We also looked at different actor categories to see if there were different patterns. This process leads to the most plausible explanations of how contextual factors influence the intervention implementation and activate mechanisms among individuals, resulting in observed outcomes. Finally, we visualised the ICAMO configurations using causal loop diagrams [47] to show complex interactions within and between them [40,46,48].

## 3. Results

### 3.1. Socio-Demographic Characteristics of Study Participants

The socio-demographic characteristics of study participants are summarised in Table 5.

### 3.2. ICAMO Configurations

This section is divided into three sub-sections corresponding to the three levels of the IPT: the PHA office level, the PHA–DHMT interface, and the district level. It presents the ICAMO components derived from empirical data, used to refine or reformulate the initial ICAMOc. While presented separately, these configurations are interconnected, as shown in the causal loop diagrams.

#### 3.2.1. At the PHA Office Level

**Intervention**. Data indicated an inadequate recruitment and training of PHA staff, low-quality meetings of PHA staff, and a lack of assignment of PHA staff to specific health districts (HDs) and their individual evaluation.

*Inadequate recruitment of PHA staff*: During the interviews, some PHA staff noted that the recruitment of new PHA staff no longer included competitive exams and often overlooked competence criteria. This issue was also highlighted in a 2020 report of an independent assessment of the capacity of nine PHA offices, including Kasai Central.


*“Unlike the recruitments resulting from the 2015 reform, which followed strict hiring procedures (call for applications, test taking, publication of results and retention of the best candidates, then submission of names to the supervisory authority for recruitment approval), subsequent recruitments were made by assignment, without any prior assessment of the new recruits’ abilities and aptitudes, which also prevented any follow-up on the latter’s background.”*
[Report of capacity assessment for 9 PHA in the DRC, 2021]

*Inadequate training of PHA staff*: PHA staff require both methodological and thematical training. The former focuses on technical support approaches, while the latter is related to technical support content. The interviews of PHA staff indicated that only one methodological training was held in 2015, and the PHA staff recruited afterwards missed it. Instead, they received sporadic, unpredictable thematic training, mainly from disease control programs.


*“We, who started with the PHA office, benefited from training on supporting health district and other subjects related to the PHA reform at the time. However, this training has no longer been provided since then.”*
[KII_8_, PHA]

*Low-quality meetings of PHA staff:* These meetings included (de)briefing sessions before and after field visits and working group meetings. They were regularly held, as shown by the review of PHA annual reports (e.g., 79% and 100% of working group meetings in 2021 and 2022, respectively). However, the interviews indicated low PHA staff attendance due to scheduling conflicts and a lack of enforcement. Additionally, poor information sharing before meetings limited the quality of discussions.


*“Working groups are like think tanks, but we often fail to share information adequately […]. We presented fresh information during meetings, but people didn’t have enough time to analyse it thoroughly and produce their best ideas. This prevented them from reflecting and contributing effectively.”*
[KII_6_, PHA]

*Assignment of PHA staff to specific HDs:* During interviews, most PHA staff reported that they were not assigned to specific HDs for an extended period. Instead, they were changed at each visit, preventing them from developing a deep understanding of the HDs, building relationships with DHMT members, and feeling accountable for the performance of the HDs they supported. The terms of reference for the reviewed technical visits corroborated this information.


*“The ideal is to keep the PHA staff in the same health district for a period of time. We have not done this for 3 or 4 years now [...], which means that this quarter you may be sent to one district, the next quarter to another, and so on.”*
[KII_3_, PHA]

*The individual evaluation of PHA staff:* During the interviews, PHA staff noted that individual evaluations could enhance competence by boosting motivation and accountability. However, they expressed concern about the lack of such evaluations at the PHA office, where only departmental evaluations were conducted under the performance-based financing (PBF) scheme. We found no documents reporting individual evaluations of PHA staff at the PHA office, confirming the interviews’ findings.


*“There is no individual evaluation *[of PHA staff]*, but they *[PBF scheme team]* evaluate the performance of departments at the PHA office based on their outputs […]. Because these evaluations involve the entire office, some members are less engaged because they think others will handle it. The situation might be different if the evaluations were done individually, as everyone would feel accountable.”*
[KII_4_, PHA]

**Context**. Contextual factors comprised insufficient financial resources, adverse political interference, and a dominant bureaucratic culture.

*Insufficient financial resources:* All data sources reported limited financial resources as a major constraint on technical support activities at the PHA office. This was due to insufficient domestic funding and a heavy reliance on external aid. The review of the annual reports of the PHA office showed that domestic funding accounted for 29% of the PHA’s working budget from 2018 to 2022, while external aid constituted 49% of the budget during the same period. However, this aid was fragmented and poorly coordinated. To address this, the PHA office signed a *“single contract”* (a virtual basket fund for all funds allocated to the PHA office) with funding agencies, but commitments were not fully met. In 2022, only 31% of the funding goals were achieved, according to the annual report of the PHA office. An excerpt from the interviews confirms this information in the following terms:


*“We are operating under the single contract, but upon evaluating the engagements of various parties, we have observed that the *[technical and financial]* partners are failing to fulfil even 50% of their obligations.”*
[KII_3_, PHA]

*Adverse political interference*: Most PHA staff interviewed noted increased political interference in the PHA office, undermining leadership and limiting decision spaces, especially regarding human resource management. Examples included overstaffing due to inadequate recruitment. In 2022, the PHA office employed 85 staff instead of the 75 required. A 2020 report of an independent assessment of nine PHA offices highlighted a similar trend.


*“Compared to the maximum set by the General Secretariat for Health, there is a surplus of 15 people in the organisation chart of the PHA office in Kasai Central.”*
[Report of capacity assessment for 9 PHA in the DRC, 2021]

Participants also reported that top PHA managers could not sanction incompetent or misconducted staff due to political protection. The PHA office’s 2021 and 2022 operational plans highlighted the *“poor implementation of proposed sanctions”* as a key issue.

*A dominant bureaucratic culture:* During the interviews, some PHA staff reported a tendency to over-compartmentalise departments within the PHA office, resulting in insufficient information sharing. Even flexible (ad hoc) bodies, such as working groups, tended to closely align with the PHA office departments they were led by, strengthening the bureaucratic culture they were supposed to mitigate.


*“We don’t really feel that information flows easily between the departments [of the PHA office] [...]. There’s a dysfunction; people are a bit compartmentalised at one point.”*
[KII_6_, PHA]

**Mechanisms.** Four mechanisms were identified at this level: motivation, self-efficacy, reflexivity, and a sense of accountability of PHA staff.

*Motivation:* Most PHA staff expressed extrinsic motivation, which was negatively impacted by insufficient funding for training and salary, poor working conditions, and inadequate career management.


*“My motivation for working at the PHA office is primarily financial because people who work must be paid and given the right conditions to do a better job, including training and other conditions, such as infrastructure and internet connection.”*
[KII1, PHA]

*Self-efficacy:* Almost all PHA staff members claimed high self-efficacy in supporting DHMTs, attributing it to their background (master’s degrees in public health and/or extensive work experience in HDs) and hands-on training (formal and informal).


*“I have gained a lot of experience working in the health districts and the provincial health administration, and I have also done a lot of on-the-job training. With this experience, I can provide effective technical support to the health districts.”*
[KII_8_, PHA]

However, some ‘Senior’ PHA staff members raised concerns about the self-efficacy of newly recruited and ill-trained PHA staff members.

*Reflexivity:* No meetings were observed during data collection, but a review of PHA meeting minutes showed limited details on discussions about practices and performance. Meetings emphasized recommendations over lessons learned. As noted above, participants raised concerns about the lack of deep reflection in working group meetings.

*Sense of accountability:* Most PHA staff reported that not being assigned to specific HDs made it difficult to evaluate their performance based on HD outcomes and reduced their accountability for the results of their support.


*“We need to attach each PHA staff member to one or two districts to support each so that he *[or she]* is accountable for the performance of the district because if the district doesn’t improve, the PHA staff member will also have his *[or her]* share of responsibility.”*
[KII_8_, PHA]

**Outcome.** Data indicated mixed competencies among PHA staff. While most PHA staff had positive perceptions of their competencies, some expressed concerns about the sub-optimal competencies of certain colleagues due to inadequate recruitment and training.


*“Some PHA staff members do not have the necessary competencies […], and this is compounded by the lack of adequate training at the PHA office.”*
[KII_6_, PHA]


**ICAMO configurations**


Based on the findings above, we formulated three ICAMOcs. The initial ICAMOcs #1a and #1b were not confirmed and were reformulated into a negative statement (ICAMO #1). The initial ICAMOc #2a was not confirmed, but its oppositional ICAMOc #2b was confirmed and refined (ICAMO #2). Additionally, one new ICAMOc emerged from empirical data (ICAMO #3).


**
*ICAMOc #1. Inadequate recruitment and training of PHA staff*
**



*Inadequate recruitment [I] of new PHA staff due to political interference [C] and their lack of proper training [I] due to inadequate financial resources at the PHA office [C] lowers the motivation [M] and self-efficacy [M] of PHA staff [A], resulting in sub-optimal technical, relational, and facilitation competencies [O].*



**
*ICAMOc #2. Low-quality meetings of PHA staff*
**



*A predominant hierarchical culture hinders information sharing within the PHA office [C], which reduces the quality of meetings [I] and prevents PHA staff [A] from effectively reflecting and learning from their experiences [M], contributing to inadequate capacity development [O].*



**
*ICAMOc #3. Lack of individual evaluation of PHA staff*
**



*In a context of political interference and inadequate human resource management at the PHA office [C], the lack of individual evaluation [I] and the absence of long-term assignments of PHA staff to specific health districts [I] reduces their sense of accountability [O] and lowers their motivation [M] to improve their competencies [O].*


The three ICAMOc at the PHA office level are visualised in Figure 2.

#### 3.2.2. At the Interface Between the PHA Staff and DHMT Members

**Intervention**. We found that technical support did not fully meet the features of effective technical support hypothesised in our IPT: being personalised and need-driven, problem-solving-centred, reflection-stimulating, comprehensive, and regular.

*PBF-driven support:* The survey indicated that most participants (100% in Bunkonde and 88% in Katoka) indicated that technical support was based on their needs, which mainly referred to weaknesses identified during previous PBF evaluations rather than their actual professional development needs. Half of them reported not being involved in identifying these needs or in drafting formal technical support plans agreed upon with PHA staff for a specific period. Despite this, interviews revealed that some PHA staff remained flexible in addressing issues raised by DHMT members during support visits, as noted by one participant.


*“There are times when they *[PHA staff]* arrive, I express the needs and describe the issue causing me problems. They also take the opportunity to brief on that issue.”*
[KII_13_, DHMT]

*Insufficiently comprehensive support:* Two types of technical support were identified: integrated or whole-system support (covering broad health system aspects and provided by PHA staff) and vertical or specific support (conducted by disease control program staff and focused on specific programme-related issues). Both document review and interviews indicated poor coordination between the less-funded integrated support and the better-funded vertical support, which led to more frequent vertical visits.


*“There are more vertical support visits *[than integrated support visits]* because disease control programs are better financed by donors. But the consequence is that technical support under these conditions is not comprehensive because these *[vertical]* support visits only deal with specific issues related to their *[disease control]* programs.”*
[KII_5_, PHA]


*“Supervision visits in the health districts exceeded targets (194%) because the vertical programs and PHA office did not have a joint agenda, resulting in visits at two paces: quarterly for some and monthly for others.”*
[Annual report of the PHA office 2019]

*Problem-solving-centred support:* All DHMT members surveyed (100%) in both HDs reported that technical support focused on solving problems, particularly those from PBF evaluations. However, the document review showed that there was no structured problem-solving approach. During interviews, DHMT members explained that PHA staff used various problem-solving approaches, including information sharing to enhance knowledge, raising awareness when dealing with inappropriate attitudes, and demonstrating tasks to improve skills.


*“We sometimes conduct joint supervision during technical support [...]. They *[PHA staff]* act as observers; let us take the lead in supervising and provide feedback at the end on what we did well and what needs improvement.”*
[KII_13_, DHMT]

*Reflection-stimulating support:* Most DHMT members surveyed (88% in Katoka and 100% in Bunkonde) felt that PHA staff stimulated reflection on their practices and performance, and all (100% in both HDs) found their feedback helpful. Interviews corroborate this, as DHMT members noted that the problem-solving process, especially root cause analysis, encouraged reflection. However, this detailed analysis was rarely documented in the technical support reports reviewed, which mostly focused on strengths, weaknesses, and recommendations presented in a linear format.

*(Ir)regular support:* Data indicated variable regularity of technical support visits depending on the type of technical support (integrated versus vertical) and the funding status of the HD (PBF versus non-PBF districts). The survey indicated that three-fourths (75%) of DHMT members in both districts noted that integrated technical support visits were irregular and too brief. The document review and interviews confirmed this and showed a mixed pattern of integrated visits: HDs with funding partners, like PBF districts, received more frequent visits, while *“orphan HDs”* without funders had fewer.


*“We are moving at two speeds. Some health districts are funded, and others are not. We regularly visited funded districts, and there are still problems with funding to cover all districts of the province.”*
[KII4, PHA]

Both document review and interviews indicated that these visits lasted 4 to 8 days, including travel, but PHA staff found this duration insufficient for effective skill transfer during integrated supervision visits. Vertical technical support visits were more regular than integrated ones.

Furthermore, participants reported instances of misconduct that led to shorter and irregular technical visits to HDs. This involved *“rushed technical support visits”* that did not adhere to the scheduled duration and *“fictitious technical support visits”*, where PHA staff received technical support funds without actually visiting HDs.

**Context**. The main contextual factors at this level included the PBF scheme, the suboptimal administrative integration of the disease control programs within the PHA office, and a mixed learning environment.

*The PBF scheme:* The review of various documents at provincial and district levels indicated that the PHA office and 12 HDs have been operating under a World Bank-funded PBF scheme to reduce stunting in children under five since 2021. The program focuses on health service improvement, strategic purchasing, and capacity building, with financial incentives to facilities tied to performance [49]. During the interviews, participants valued the incentives provided by the PBF scheme, but they raised concerns about delayed fund disbursement.


*“We signed a contract with the financial partners, who were supposed to pay us based on our health district’s performance at the end of every quarter. However, as we speak, we have not received these funds for two and a half quarters.”*
[KII_15_, DHMT]

*Sub-optimal administrative integration of the disease control programs within the PHA office:* Fragmented external aid, mainly through disease control programs, led to poor administrative integration of these programs within the PHA office. Integrating disease control programs involves embedding their activities into general health services [50,51]. This includes administrative integration, where multi-function health managers decide about vertical programs, and operational integration, where multi-function providers carry out program activities. Both types of integration are crucial for mutual benefit [50,52].

During interviews, some PHA staff noted that disease control programs were more accountable to national directorates than the PHA office. This lack of coordination resulted in inefficient use of technical support funds.


*“The provincial disease control programs offices do not feel much connection with the technical support department. Even if they conduct vertical supervision visits, they don’t coordinate with the technical support department.”*
[KII_6_, PHA]

*Mixed learning environment*: Most interviewed DHMT members reported experiencing a mixed learning environment, shaped by the attitudes of PHA staff. Most PHA staff reported adopting a supportive attitude, but some DHMT members felt certain PHA staff displayed a dominant, hierarchical, and judgmental attitude.

**Mechanisms.** The mechanisms operating at this level were motivation, the perceived relevance of the support by DHMT members, the perceived credibility of PHA staff among DHMT members, psychological safety, mutual trust and openness, the reflexivity within the DHMT, and the sense of accountability.

*Motivation*: Like PHA staff, DHMT members were mainly extrinsically motivated. Despite the reported delays in performance bonus payments, the PBF scheme was found to be a motivating factor.


*“You know that people aren’t well paid. So, this project *[PBF scheme]* motivates people to work because they know that if they don’t work well, they will not be rated highly, so they will not get much money.”*
[KII_17_, DHMT]

*Perceived relevance of the support:* Interviews indicated that DHMT members valued the pedagogical nature of technical support visits, especially those addressing issues from previous PBF evaluations, as these were seen as highly relevant due to the financial incentives tied to improved performance.


*“When there is a technical support visit, it is in our interest to actively participate because we gain from it [...] as it helps us perform our duties better and have a good score *[during the upcoming PBF evaluation]*.”*
[KII_17_, DHMT]

*Perceived credibility of the PHA staff:* DHMT members appreciated the credibility of the PHA staff based on their competencies and attitudes. All DHMT members surveyed (100%) in both HDs positively perceived the PHA staff’s technical and facilitation competencies. However, over a third of them agreed (25%) or were neutral (13%) about hierarchical or fault-finding attitudes, reflecting the mixed relational competencies of PHA staff. Supportive and competent staff were seen as more credible, while less competent or judgmental staff were considered less credible. This is echoed in the following quotation:


*“When PHA staff bring their expertise and provide helpful support, it is in our interest to stay connected with them. But, when other PHA staff come and only criticise without offering any helpful input, it can be frustrating. Some DHMT members may even request permission to be absent, citing other reasons, just for not participating *[in technical support visit]* when a particular PHA staff member arrives because they know that person is not helpful.”*
[KII_13_, DHMT]

*Psychological safety:* DHMT members’ psychological safety depended on PHA staff attitudes. Interviews indicated that supportive staff increased psychological safety, while hierarchical or judgmental staff decreased it. In the latter case, DHMT members adopted a conformist approach, agreeing with PHA staff to expedite visits and minimize interaction.


*“But there are other PHA staff who act like inspectors and condemn everything […]. So, we begin to say yes to everything he says; we cannot express ourselves anymore.”*
[KII_13_, DHMT]

*Mutual trust and openness:* During the interviews, participants at the provincial and district levels underscored the importance of trust and openness between PHA staff and DHMT members.


*“If the supervisees *[DHMT members]* do not trust the supervisor *[PHA staff]*, they will not open up to him *[or her]* to talk about their problems and look for solutions.”*
[KII_7_, PHA]


*“You know, in any relationship, trust is very important, not just for technical support, but also within the management team and our families and couples [laughs]. It makes the relationship easier. If you trust someone, you tell him [her] your problems, and you help each other.”*
[KII_15_, DHMT]

*Reflexivity:* DHMT members found problem analysis during technical support visits helpful for reflecting on and learning from their practices.


*“When we reflect, we look for the causes of problems and solutions. If we find the solutions and apply them, we solve the problems, improve the team’s performance, and we learn too.”*
[KII_15_, PHA]

However, we could not observe these sessions during data collection, and the reviewed technical support reports lacked detailed information on the reflection process.

*Sense of accountability:* Interviews indicated that DHMT members felt a high sense of accountability due to the PBF scheme and performance rankings during annual review meetings. Within both DHMTs, each member was responsible for specific PBF indicators and was regularly evaluated by the HD head before formal PBF assessments.


*“Each DHMT member is responsible for some *[PBF]* indicators; he *[she]* has to monitor and ensure that the assigned indicators evolve in the right direction.”*
[KII_13_, DHMT]

**Outcome**. Our analysis indicated the management capacity of both DHMTs to be average.

Qualitatively, most DHMT members had positive perceptions of their management competencies, crediting technical support for improving their competencies in supervision, health information, and drug management. However, some PHA staff felt technical support had a limited impact due to political interference, such as unqualified DHMT appointments and untimely reassignments.

The quantitative assessment showed the overall management capacity for both DHMTs to be average: 70% in Katoka and 66% in Bunkonde (Figure 3). Their strongest functions were planning, monitoring and evaluation, health information, and drug management, with Bunkonde excelling in financial management.


**ICAMO configurations**


Based on the findings above, we formulated five ICAMOcs. The initial ICAMOc #3a was not confirmed and was reformulated as ICAMOc #4. The initial ICAMOc #3b was confirmed and refined into ICAMOc #5. Both initial ICAMOcs #4a and #4b were confirmed and refined into ICAMOcs #6a and #6b. The initial ICAMOc #5 was also confirmed and refined as ICAMOc #7. Additionally, ICAMOc #8 emerged from empirical data.


**
*ICAMOc #4. PBF-driven support*
**



*In the context of the PBF scheme [C], basing technical support on issues identified during performance evaluations [I] increases its perceived relevance [M] among DHMT members [A] because of extrinsic motivation [M] due to expected financial incentives linked to performance improvement. This encourages active participation in the technical support process, leading to improved competencies [O].*



**
*ICAMOc #5. Insufficiently comprehensive support*
**



*Inadequate administrative integration of disease control programs, coupled with PBF targeting mostly disease control indicators [C], hampers the comprehensiveness of technical support [I]. This may reduce the perceived relevance of the support [M] among DHMT members [A], negatively impacting their participation and competencies [O].*



**
*ICAMOc #6. Problem-solving support*
**


***#6a.*** *Effective problem-solving support [I], especially related to the PBF framework [C], in a conducive learning environment [C], enhances the perceived credibility of PHA staff [M] among DHMT members [A]. This fosters psychological safety, trust, and openness [M], encouraging active participation in the technical support process and resulting in improved competencies [O].*

***#6b.*** *Conversely, a hierarchical attitude from PHA staff [C] hinders psychological safety [M] for DHMT members [A], leading to a conformist attitude [M] that restricts interactions and hampers learning and capacity development [O].*


**
*ICAMOc #7. Reflection-stimulating support*
**



*Meaningful reflection and constructive feedback [I] in a conducive learning environment [C] enable psychological safety and reflexive learning [M] among DHMT members [A], leading to improved competencies [O].*



**
*ICAMOc #8. (Ir)regular support*
**



*In the context of the PBF scheme [C], regular technical support visits after performance evaluations [I] enhance the sense of accountability [M] and extrinsic motivation [M] of DHMT members [A], leading to active participation and learning that improve competencies and performance [O].*


The five ICAMOcs at the interface between PHA staff and DHMT members are visualised in Figure 4.

#### 3.2.3. At the District Level

**Intervention**. Interviews indicated that management practices at the district level were more PBF-oriented. DHMTs tended to prioritise purchased management activities because of financial incentives linked to improved performance. The review of the working plans, supervision reports, and meeting minutes of DHMT members corroborated this finding.


*“We have the performance criteria for the main functions of the DHMT. At the end of each quarter, we are evaluated against these criteria. Before this evaluation, we worked extensively on these main functions to improve our performance.”*
[KII_14_, DHMT]

**Context**. DHMTs encountered the same contextual factors as the PHA office, including the PBF scheme, hierarchical culture, and political interference. During interviews, DHMT members reported that the hierarchical culture and political interference at the PHA office negatively impacted their leadership and decision spaces, particularly in managing human and financial resources.


*“There are also political influences; the public administration becomes more politicised, and managing people within this context is really challenging.”*
[KII_18_, DHMT]

**Mechanisms**. The operating mechanisms at this level were extrinsic motivation (detailed above), self-efficacy, and the perceived autonomy of DHMT members.

*Self-efficacy:* Some DHMT members felt technical support enhanced their competencies, boosting confidence in managing health information, drug management, and supervising health workers. This is explained by a PHA staff member:


*“For instance, when it comes to data, thanks to the technical support, I can now analyse the quality of the data on DHIS2, detect deviations from established rules, make corrections and independently manage the process without encountering challenges”.*
[KII_16_, DHMT]

*Perceived autonomy*. Despite their self-reported (extrinsic) motivation and self-efficacy, DHMT members complain about the low perceived autonomy for decision-making regarding human and financial resource management due to a strong hierarchical culture and political interference. This is expressed in the following quotation:


*“Our decision-making authority regarding health worker assignment and sanction is limited as official texts do not grant us this power. This limitation makes it challenging to deploy personnel effectively.”*
[KII_18_, DHMT]

**Outcome**. Regarding the performance of health districts, respondents noted that technical support positively impacted HD performance within the PBF scheme. They noted, for example, the increased accessibility and utilisation of healthcare services and improved quality of care. The following excerpt from the interviews explains this:


*“In most cases, there is a positive trend *[of the performance of the health districts]*, which leads us to believe our *[technical]* support has brought about effects in the right direction.”*
[KII_6_, PHA]

Based on quantitative routine data collected from DHIS2 software from 2018 to 2022, both HDs have shown good performance, with a score of 84% for Katoka HD and 80% for Bunkonde HD in 2022 (Figure 5).


**ICAMO configuration**


The initial ICAMOc #6 was confirmed and refined into ICAMOC #9 as follows:***ICAMOc #9. PBF-oriented management practices***


*In the context of the PBF scheme [C], enhanced management capacities of DHMT members [A] increase their self-efficacy and motivation [M] to implement management activities targeting key performance indicators [I], thereby improving health district performance [O]. However, weak leadership and narrow decision spaces due to a hierarchical culture and political interference [C] undermine these improvements [O] because of the low perceived autonomy of DHMT members [M].*


The ICAMOc at the district level is visualised in Figure 6

#### 3.2.4. Putting All Together: The Refined Program Theory

We connected ICAMO configurations into an overall refined IPT (Box 1).

Box 1The refined program theory.In resource-limited settings, an externally funded PBF scheme brings up resources and serves as a motivation and accountability framework for health workers at provincial and district levels. This program directs technical support content towards purchased activities, which DHMT members find highly relevant due to the financial incentives tied to performance improvement. When delivered by competent, credible, and trustworthy PHA staff in a supportive learning environment, this support fosters active participation, reflexive learning, and enhanced competencies. Improved DHMT competencies boost their self-efficacy in performing management activities targeting key performance indicators, thereby enhancing health district performance. However, these gains can be offset by challenging contextual factors, such as poor leadership, limited decision space, inadequate human resource management, and a hierarchical culture.

## 4. Discussion

The study found that PHA staff’s technical support to DHMT members in the DRC is sub-optimal. It is influenced by various contextual factors, triggering various mechanisms that limit the achievement of expected outcomes.

### 4.1. Addressing the Elephant in the Room

We found that the sub-optimal implementation of technical support was mainly due to contextual factors, such as resource issues (insufficient domestic funding and fragmented external aid), political interference, weak leadership, and the hierarchical organizational culture. Although these issues are well-documented in health policy documents [14,17,18,53,54,55], they often go unaddressed in health interventions due to their complex and structural nature. This discourages health actors from tackling them at individual or organizational levels, leading to various survival strategies (corruption, clientelism, peddling influence, etc.) that undermine intervention effectiveness. These coping strategies become embedded as *“practical norms”* in organisational culture and overshadow fundamental issues, hence the metaphor of *the elephant in the room.*

To address these issues, it is essential to re-attract attention to *‘the elephant’* and then remove it from *‘the room’*. This requires sustained political will and coordinated actions to strengthen the MoH’s stewardship at all levels. These actions involve tackling the politicization of public administration and its corollaries, such as clientelism and corruption, improving the management of available resources while gradually increasing the national health budget, and establishing a robust accountability framework. Such measures can restore MoH credibility among external funders and encourage them to align with national priorities [56]. This process is complex, requiring a reliable system for monitoring, evaluation, and learning to adapt actions based on emerging behaviours and evolving contexts.

### 4.2. The PBF Program: A Valued but Not Sustained Game-Changer

In a context of insufficient domestic funding, the PBF scheme was found to be a valued alternative funding source that significantly influenced the implementation of technical support in various ways. First, the PBF scheme funds regular technical support visits to HDs, with per diems offering financial incentives for PHA staff. However, in the DRC, where health workers’ remuneration is low, per diem can lead to negative effects that undermine the health system’s efficiency [57]. These effects include a culture of per diem hunting, which can lead to various misconducts such as corruption, clientelism, or fictitious or rushed technical visits. Second, since the PBF scheme only covers some HDs, it creates a *double-speed* support scenario, with funded districts receiving frequent visits and non-funded districts becoming *orphan districts*. This practice perpetuates healthcare inequity and impedes progress towards universal health coverage, as the ultimate goal of technical support is to improve the health status of the population.

Third, the PBF scheme significantly influences technical support content by over-focusing on issues identified during performance evaluations, which focus on the inherently narrow priorities of the PBF schemes. DHMT members valued this support due to financial incentives linked to performance improvements. However, this *cream-skimming* [58] reduces the comprehensiveness of support by creating tunnel vision, with more focus on funded indicators, neglecting other important needs of DHMT members. Furthermore, over time, financial incentives may lead to diminishing marginal returns, as the same level of incentives may no longer proportionally boost motivation or performance if other non-financial motivators are not in place [59]. Performance-contingent monetary rewards have been shown to undermine intrinsic motivation or the internalisation of extrinsic motivation among autonomously motivated health workers in a control-and-command environment [59,60]. Lastly, the PBF program’s reliance on external funding raises concerns about the sustainability of its outcomes in the context of insufficient domestic health funding [61,62]. With the program ending in 2026, all things being equal, there is a high risk of reverting to previous performance levels.

### 4.3. Framing the Refined PT into Existing Substantive Theories

The refined program theory highlights the role of the PBF scheme in improving the performance of health districts through financial incentives. The PBF scheme aims at aligning the behaviour of health workers with the funder’s interests through contracts and monitoring, as described by principal-agent theory [63]. Rational choice theory suggests that health workers’ behaviour may also be driven by personal utility maximization rather than the funder’s monitoring system [64]. A well-known issue with PBF schemes is the *cream-skimming effect* [58], which can be linked to the goal displacement theory, where the focus shifts from original objectives to incentivized goals [65]. The PBF-oriented support and management practices evidenced in this study illustrate this goal displacement theory.

The refined program theory underscores that DHMT members found PBF-driven technical support highly relevant due to financial incentives linked to performance. While non-financial incentives, like intrinsic motivation or professional growth, could also enhance perceived relevance, empirical data did not support this in the context of the study sites. The perceived relevance of technical support aligns with Knowles’ adult learning theory, which emphasizes the importance of training relevance in the learning process [23], and the health belief model, which underscores perceived benefits in adopting health behaviours [66].

The refined program theory stresses challenging contextual factors, including political interference, hierarchical culture, poor leadership and narrow decision spaces. These reflect organizational politics, where power is used for self-interest [67,68,69]. It can negatively impact efficiency and effectiveness, job satisfaction, and commitment. It can also increase job stress, intention to leave, and workplace deviance, such as corruption and clientelism, as seen in the study [67,69].

### 4.4. Implications for Practice

The study highlights the need to address contextual challenges to improve technical support for DHMTs. While long-term efforts are crucial, short- and medium-term actions are also necessary. First, structured training for PHA staff is key to enhancing their technical, facilitation, and relational competencies. This training should cover both methodological aspects (i.e., technical support approaches) and thematic areas (i.e., DHMT functions). An action-learning approach, blending short classroom sessions with field mentoring or coaching, could be particularly effective [37]. Second, PHA meetings should be improved as genuine learning spaces through better preparation, scheduling, and information sharing. Third, PHA staff should be assigned to specific HDs for at least a year to foster accountability and trust with DHMT members. Fourth, a minimum level of the coordination of technical support actors and funding is necessary to ensure synergy and consistent support across HDs. Finally, addressing motivation—through an optimal mix of financial and non-financial incentives like supportive leadership, a positive work environment, teamwork, recognition, and career development—is important for sustained engagement and performance.

### 4.5. Study Rigour, Trustworthiness, and Limitations

In realist studies, rigor refers to the trustworthiness of the data and their sources, the quality of the analysis and the coherence of the program theory [70]. In this study, rigor was ensured by triangulating data from various sources and developing ICAMO configurations rather than categorizing ICAMO components thematically [71]. We enhanced program theory coherence by exploring rival theories and aligning our refined program theory with substantive theories. Feedback from supervisors and experts in realist evaluation and health system research further strengthened the analysis. Additionally, we adhered to reporting standards for RE using the RAMESES II checklist (Appendix A) [72].

This study has limitations. The document review may not be exhaustive due to inadequate archiving at PHA and HD offices, and routine data from the National Health Information System (DHIS2) may be of poor quality. Using performance rankings based on these routine data to select the study province could also introduce a selection bias. There may also be social desirability bias, with participants potentially withholding negative views due to the hierarchical culture. Triangulating data from multiple sources helped mitigate these issues. Additionally, data collection guided by the IPT risks tunnel vision, where only corroborating elements are sought [73]. To minimize this, we continuously sought feedback from supervisors throughout the research.

## 5. Conclusions

This study used the RE approach to test the IPT of technical support provided by PHA staff to DHMT members. Results showed mixed outcomes due to sub-optimal implementation influenced by contextual challenges related to resource availability and political and organizational environments. The PBF scheme helped balance these challenges by providing resources and boosting health actors’ motivation, but its external funding feature raises sustainability concerns. The study underscores the need for strong political will and coordinated actions to address these contextual challenges and ensure effective technical support for DHMTs.

## Figures and Tables

**Figure 1 ijerph-21-01646-f001:**
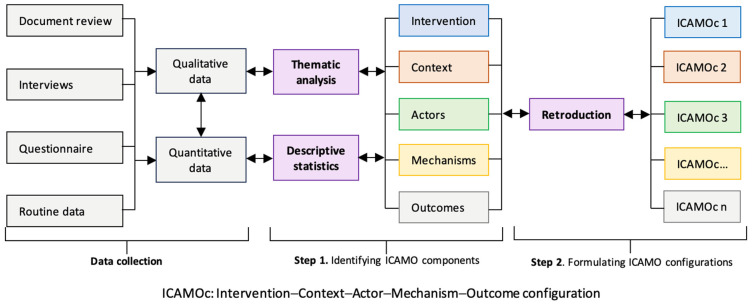
Data analysis process.

**Figure 2 ijerph-21-01646-f002:**
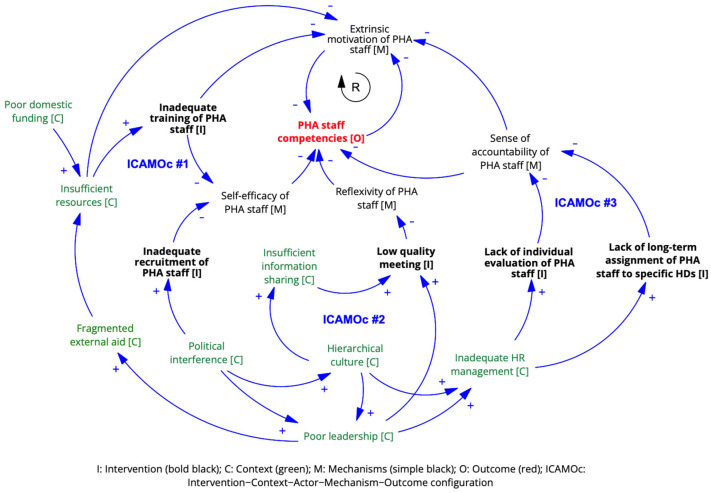
ICAMOc at the PHA office level.

**Figure 3 ijerph-21-01646-f003:**
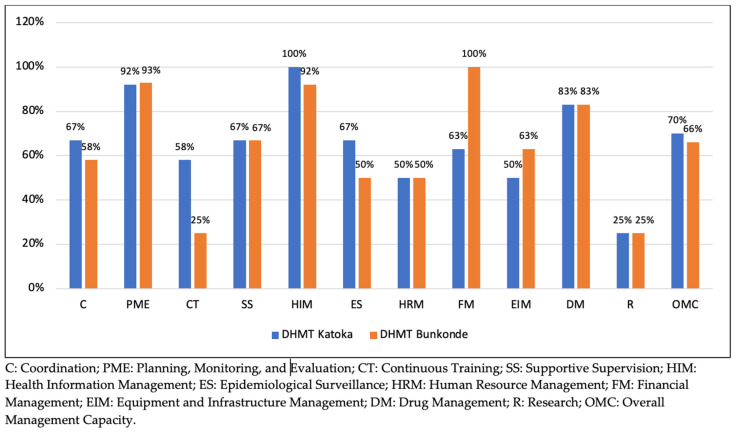
Management capacities of DHMTs.

**Figure 4 ijerph-21-01646-f004:**
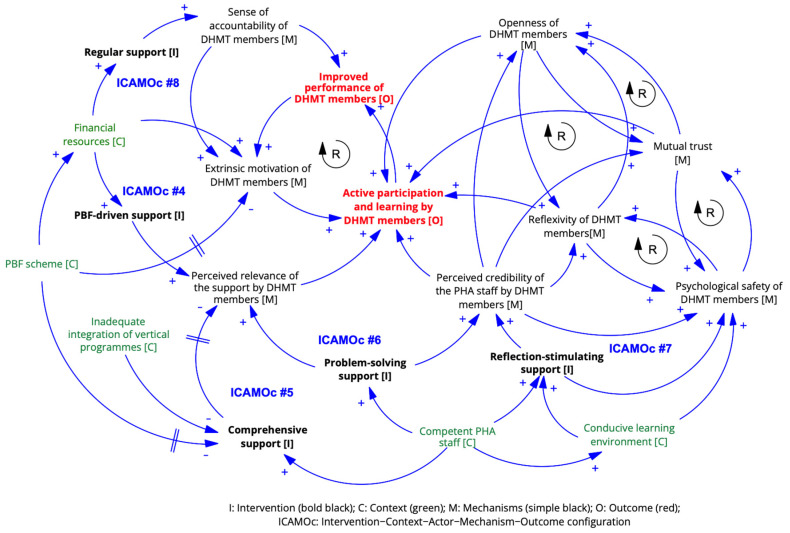
ICAMO configuration at the interface between PHA staff and DHMT members.

**Figure 5 ijerph-21-01646-f005:**
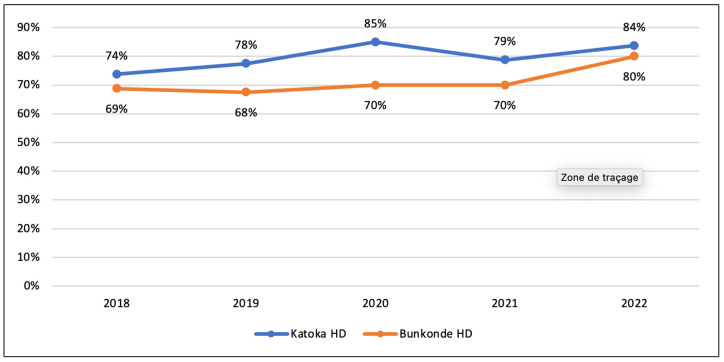
Performance of health districts.

**Figure 6 ijerph-21-01646-f006:**
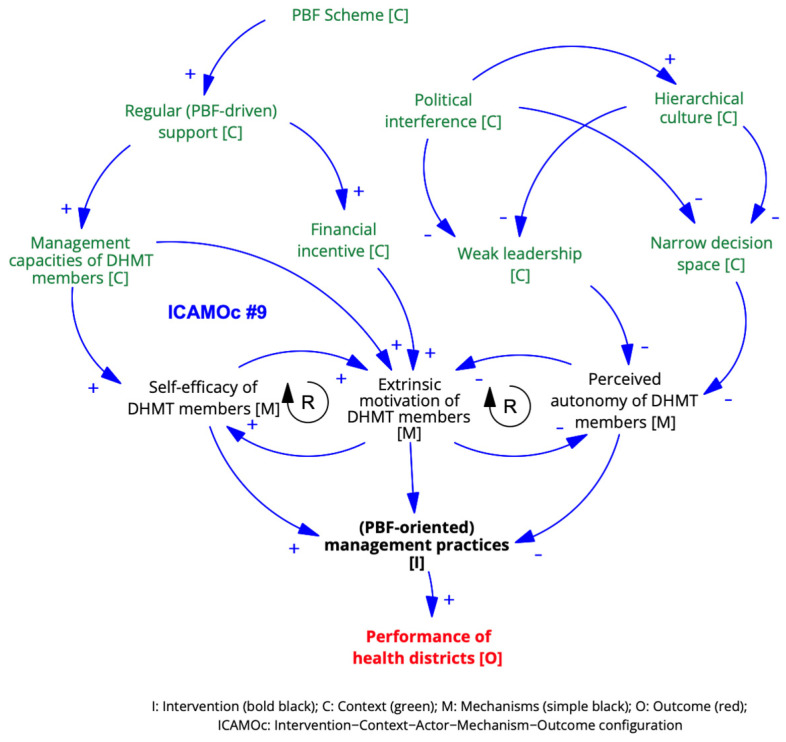
ICAMO configuration at the district level.

**Table 1 ijerph-21-01646-t001:** The initial program theory [21].

**At the PHA level** ICAMOc #1 ***#1a.*** *Training in management and facilitation, including relational knowledge and skills (I) targeting PHA staff (A), increases their self-efficacy (M) and motivation (M), leading to improved competencies (O) and commitment to providing technical support to DHMT members (O). A good work climate, promotion of positive values, and provision of adequate resources (C) at the PHA office are essential.* ***#1b.*** *Unsupportive leadership in a context of inadequate resources (C) may demotivate (M) PHA staff (A) and lead to the exit of staff, reducing the number of skilled staff at the PHA office and thus jeopardising the technical support process (O).* ICAMOc #2 ***#2a.*** *Regular meetings at the PHA office to plan, evaluate, and discuss technical support issues (I) offer PHA staff (A) opportunities to share, reflect on, and learn from their field experiences, enabling psychological safety (M) among PHA cadres and contributing to reflexivity (M), which leads to improved competencies (O) through individual and collective learning on the condition of safe conversational spaces that value and respect everyone’s opinions and encourage people to speak up (C).* ***#2b.*** *Conversely, a highly hierarchical management culture (C) can create psychological unsafety (M), making PHA staff (A) hesitant to share their opinions for fear of being judged, embarrassed, or punished. This inhibits both reflexivity and learning and thus hinders the development of competencies of PHA staff (O).*
**At the interface between PHA staff and DHMT members** ICAMOc #3 ***#3a.*** *DHMT members’ involvement in identifying their own support needs and planning support visits (I) results in a positive perception of the relevance of the support received (M), encouraging their active participation in the technical support process and improving their competencies (O). This is more likely to occur in an environment conducive to learning (i.e., that is judgement-free, fault-accepting, non-threatening, and less hierarchical and where there are supportive relationships between PHA staff and DHMT members) (C).* ***#3b.*** *Conversely, vertical supervision visits by disease control program staff (I), that do not necessarily meet the needs of DHMT members (A), lead to perceptions of the irrelevance of such supervision (M), hindering professional development and ultimately resulting in less-than-optimal performance (O).* *ICAMOc #4* ***#4a.*** *DHMT members (A) are likely to participate effectively in technical support and thus improve their competencies (O) if they perceive the PHA staff as credible (M) and trustworthy (M). These positive perceptions of credibility and trustworthiness are triggered if the PHA staff has good management, facilitation, and relational skills (A), which allow them to provide effective problem-solving support (I) to DHMT members and set up a conducive learning environment that is judgement-free, fault-accepting, non-threatening, and less hierarchical and fosters supportive relationships with DHMT members (C).* ***#4b.*** *However, supervision by PHA staff members with a hierarchical attitude (I) may be perceived as less credible (M) and trustworthy (M) by the DHMT members (A), hinder their psychological safety (M), and result in weak or reluctant participation in the technical support process (O), ultimately hampering the performance of health districts (O).* *ICAMOc #5* *If competent PHA staff members stimulate meaningful reflections and provide constructive feedback (I), in a learning environment that is judgement-free, fault- accepting, non-threatening, and non-hierarchical, and where relationships between PHA staff and DHMT members are supportive (C), then DHMT members may become more reflexive (M), which contributes to individual and collective learning and ultimately improved competencies (O).*
**At the district level** * ICAMOc #6 * *If supervision (I) increases their competencies, DHMT members (A) will be more motivated to develop management initiatives to improve their health districts’ performance (O) because of higher self-efficacy (M) and perceived autonomy (M). Favourable contextual conditions include strong leadership, a supportive work environment with adequate resources, and an absence of negative political influences (C).*

**Table 2 ijerph-21-01646-t002:** Key features of the study setting.

	Kasai Central PHA	Katoka HD	Bunkonde HD
Area surface (Km^2^)	69,849	192	3600
# Population	5,529,886	158,093	141,604
# Health districts	26	NA	NA
# Health areas	458	10	14
# District hospitals	24	1	1
# District hospital beds	3103	103	89
# First-line facilities *	996	32	28
Geographic health coverage **	-	100%	88%
Epidemiological profile (leading causes of morbidity)	Malaria, acute respiratory infection, malnutrition, measles (recurrent outbreaks since 2019)	Malaria, acute respiratory infection, typhoid fever, malnutrition	Malaria, acute respiratory infection, malnutrition, simple diarrhoea
Accessibility	Accessibility challenges including poor road conditions and inadequate public transport	Easy access by road; 4 km away the PHA office	Difficult access by road; 75 km away the PHA office
Security	Isolated inter-ethnic conflicts in eight HDs	Good	Good

* In a context of weak regulation and anarchic proliferation of health facilities, this number may underreport the actual count, particularly in urban areas, as it only includes facilities compliant with the health administration’s requirements for reporting. ** The geographic health coverage is the proportion of the population living within 5 km or an hour’s walk of a health facility and having no significant geographic barriers. HD: health district; NCD: non-communicable diseases; PHA: Public Health Administration. Sources: Operational plans and annual reports of the PHA and the two HDs for 2022.

**Table 3 ijerph-21-01646-t003:** Data collection methods and tools.

Methods	Purposes	Tools	Sampling, Participants, and Size
**Document review**	Describe the actual implementation of technical support and contextual factors	Excel sheet and field notes	All relevant documents (annual plans and reports, reports and terms of reference and technical support visits, minutes of various meetings, evaluation reports, etc.) available at the PHA and HD offices from 2018 to 2022 were screened
	Assess the DHMT management capacities (outcome)	Scorecard with score ranging from one (very poor) to four (very good) for each item from core management functions assigned to DHMTs in the DRC (Appendix A)	See Appendix A for the list of documents reviewed and management functions evaluated
**Semi-structured interviews** (conducted by first author, face-to-face, in French, audio-recorded, average duration: 42,5 min)	Gather data related to the implementation features, contextual factors, mechanisms, and perceived outcomes	Interview guides developed on the basis of the IPT and piloted (Appendix A)	Purposively selected PHA staff (n = 8) and DHMT members (n = 10; 5 in each HD)
**Questionnaire** (self-administered)	Gather DHMT members’ perceptions about the process and outcomes of the technical support	Questionnaire with a 5-point Likert scale (ranging from 1 = strongly disagree to 5 = strongly agree) developed on the basis of the IPT and piloted (Appendix A)	Exhaustive sampling of DHMT members (n = 16 out of 26 members (61.5%) of the two DHMTs) present at the HD offices during our visits
**Routine data** from DHIS2	Assess the performance of the selected HDs from 2018 to 2022 (outcome)	Scorecard with score ranging from one (very poor) to four (very good) for each of ten indicators from the monitoring and evaluation framework of the National Health Development Plan 2019–2022 (Appendix A)	See Appendix A for the list of indicators used and their operational definitions

**Table 5 ijerph-21-01646-t005:** Socio-demographic characteristics of participants.

Participants	Interviews (n = 18)	Questionnaire (n = 16)
Affiliation		
PHA office	8	-
Katoka HD	5	8
Bunkonde HD	5	8
Age	Mean = 51 ± 5.5	Mean = 49 ± 9.1
<40 years	0	1
40–49 years	5	5
50–59 years	12	8
≥60 years	1	2
Gender		
Male	15	10
Female	3	6
Qualification		
Physicians	9	3
Nurses	5	9
Administrators	2	2
Undergraduates in public health	2	2
Duration in the current position	Mean = 9.5 ± 6.5	Mean = 9.6 ± 7.9
1–5 years	5	7
6–10 years	9	4
>10 years	4	5

## Data Availability

The original contributions presented in the study are included in the article/Appendix A; further inquiries can be directed to the corresponding author.

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
