# Peer review of "How Does the Context Shape the Technical Support from the Provincial Health Administration to District Health Management Teams in the Democratic Republic of Congo? A Realist Evaluation"

_ijerph, 2024, doi:10.3390/ijerph21121646_

Round 1
Reviewer 1 Report
Comments and Suggestions for Authors
This paper examines how contextual factors affect the effectiveness of technical support provided by the Provincial Health Administration to District Health Management Teams in the Democratic Republic of Congo. While the research question is significant, the current draft of the paper has several areas that require improvement:
- The method for calculating the management capacity of DHMTs and the performance of Health Districts is unclear. Rather than leaving all information in the appendix, the authors should provide a concise description of the measures and calculations in the main text. Additionally, the authors should discuss how the thresholds for "good," "average," and "low" are determined and clarify what each measure aims to capture.
- The authors should add a literature review on relevant studies, particularly focusing on theories or empirical analyses of common mechanisms that may affect the effectiveness of technical support. This would help contextualize the findings within the broader body of existing research.
- The evaluation of HD performance coverd the 2018-2022 period. The authors should include a discussion on how COVID-19 may have affected the outcome measure. The authors should also discuss how COVID-19 affected the health delivery system in DRC, as well as the interactions between the pandemic and the various ICAMOc identified in this study.
Reviewer 2 Report
Comments and Suggestions for Authors
The manuscript addressed an important issue related to health care system in the Democratic Republic of Congo. It is a well written and structured, making it engaging and understood by readers. However, I would like to highlight the following issues for your (authors’) consideration.
To being with, there appears to a discrepancy between the focus implied by the article’s title and the focus presented in methodology section. From the title ‘How does the context shape the technical support from provincial health administration to district health management teams in the Democratic Republic of Congo? A realist evaluation’ one would expect the article explore and describe how ‘context shape technical support provided by Provincial Health Administration to district health management teams. However, in Table 4 (line 181), four elements including the context, are described as factors that shape technical support or influence the outcome of the technical support.
Additionally, while thematic and descriptive statistics are reportedly used for data analysis (Figure 2, Line 195), it would be beneficial to specify which of the ICAMO elements are analysed using either of these data analysis method, or both.
In the ‘Results’ section, it is essential to give due attention to (not necessarily equal) data collected through interviews, document review, and survey. The readability of this section would be enhanced by increasing the level of triangulation. I suggest that triangulating data collected through the three methods to improve organisation and coherence.
Including many verbatim quotations in the text would be helpful, as it allows research participants to have a voice. As a qualitative researcher(s), it is essential not only present these quotes, but also to interpret their meaning.
Lastly, there is a minor correction needed in lines 171-172, where the word "intervention" appears twice. Please remove one instance.
Reviewer 3 Report
Comments and Suggestions for Authors Thank you for the opportunity to review this manuscript. The authors have conducted decent work in providing in-depth views of the current public health situation in DR Congo. The reviewer would like to congratulate the authors for their hard work and effort to conduct this research. The work is overall well constructed, and the presentation of their findings is well written. One comment I would add would be about how the contents of the study were collected. The authors roughly included the documents and the interviewees, but the current documentation does not fully contain the information to replicate this study. The authors mention the documentation bias as a limitation, but the selection criteria may also be a limitation. However, the current manuscript does not contain enough information to evaluate this.Author Response
Please see the attachment
